# Accurate and Rapid Measurement of Soil Dry Depth Using Ultrasonic Reflection Waves

Zhongwei Liang [1,2], Chunhui Zhao [1,2], Yupeng Zhang [3], Sheng Long [1,2], Jinrui Xiao [2,*] and Zhuan Zhao [4,*]

1   Guangdong Engineering Research Centre for Highly Efficient Utility of Water/Fertilizers and Solar-Energy Intelligent Irrigation, Guangzhou University, Guangzhou 510006, China; liangzhongwei@gzhu.edu.cn (Z.L.); bze284@163.com (C.Z.); hnlongs@163.com (S.L.)
2   School of Mechanical and Electrical Engineering, Guangzhou University, Guangzhou 510006, China
3   China-Ukraine Institute of Welding, Guangdong Academy of Sciences, Guangzhou 510650, China; zhangyp@gwi.gd.cn
4   School of Physics and Materials Science, Guangzhou University, Guangzhou 510006, China
*   Correspondence: meexiaojinrui@gzhu.edu.cn (J.X.); zhuan.zhao@gzhu.edu.cn (Z.Z.)

**Abstract:** Soil dry depth is a key parameter that determines soil fertility and nutrient availability, ultimately affecting crop yield and quality. However, accurately measuring the dry depth of soil has been a challenge. In this work, we propose using reflective ultrasonic waves to measure dry depth in soil. Four soil types, including clay, sandy loam, silty loam, and sandy were prepared and the feasibility of the method was demonstrated through theoretical analysis. An experimental measurement system was established to verify the consistency between ultrasonic measurements and manually measured values. Two statistics were used in Ordinary Least Squares (OLS) regression to evaluate the model fit: R-square ($R^2$) and Root mean square error (RMSE). The results indicate that the proposed method provides a higher accuracy in estimating the dry depth of sandy loam and silty loam ($R^2$ values of 0.9899 and 0.992 for sandy loam, RMSE values of 1.57% and 1.5% for silty loam) than those of the clay and sandy samples ($R^2$ values of 0.9896 and 0.9874 for clay, RMSE values of 1.66% and 1.77% for sandy). The maximum measurement errors for all the soil type predictions are below 6%; the overall accuracy was acceptable. Our findings suggest that ultrasonic measurement is an efficient and cost-effective approach for measuring soil dry depth, which could enable the precise control of irrigation water usage and the conservation of valuable water resources.

**Keywords:** soil dry depth; ultrasonic wave; measurement error




## 1. Introduction

Water is a crucial and indispensable element for the growth of plants. Appropriate water irrigation can promote the healthy growth of crops and enhance yield and quality. However, the conventional method of flood irrigation used in field production often leads to excessive irrigation and low water use efficiency [1]. Excessive irrigation can result in various problems, such as root hypoxia and root rot [2,3], which ultimately reduce crop yields. Therefore, monitoring soil moisture is essential as it is one of the most important indicators for determining soil fertility and nutrition.

The classic method of measuring soil moisture is the drying method [4], which provides a high accuracy and is independent of soil type and salinity. However, this method does not allow a repetitive measurement [5]. In addition, it is time-consuming and laborious, making it unsuitable for continuous monitoring [6,7]. In recent years, several new methods have been proposed, including the tensiometer [8,9], the neutron probe [10–12], gamma radiation [13–15], infrared remote sensing [16,17], and dielectric methods [18–21]. For instance, Masoud et al. [22] used the angular distribution of scattered neurons to measure soil moisture with small relative errors ranging from 2 to 10%. Zhao et al. [23]

used four algorithms with L-band radiometry at different angles and found that the accuracy of all the algorithms achieved their best performances at intermediate incidence angles of 40° to 45°. Calamita et al. [24] estimated soil moisture in eight different sites using Resistivimeter Syscal Junior and a portable Time domain Reflectometer (TDR), and found that the resistivity measurements were superior in inferring soil moisture spatial and temporal variability with an average RMSE of 4.4%. In general, tensiometers are cost effective and non-destructive, and they can provide continuous measurement without distressing the soil. However, tensiometers require frequent maintenance to supply water to the detector [25]. Although neutron probes and gamma radiation can provide fast and accurate soil moisture measurements at a fixed point, the equipment is expensive and the radiation may bring health risks [26,27]. Infrared remote sensing has a low penetration depth, and the response is significantly affected by environment and climate conditions [28]. Moreover, the dielectric moisture sensors (i.e., TDR, FDR, SWR sensors) are also expensive and their applicability in highly saline soils is limited.

Despite many efforts to measure soil moisture, the dry depth of soil is a more critical parameter in determining how much water and nutrients a crop can absorb. This parameter refers to the depth from the soil surface to the interface of the dry and wet soil, where dry soil has moisture content below 5% and wet soil has a moisture over 5%. The accuracy of the tools and techniques in measuring soil dry depth depends on many factors, such as soil texture, temperature, wind, and measurement methodology. Currently, the measurement of soil dry depth is poorly investigated, and limited efforts have been made to measure frozen soil depth using techniques such as ground penetrating radar [29], electrical resistivity tomography [30], or borehole thermometry [31]. Nevertheless, these methods are expensive and are not suitable for measuring soil dry depth. Developing a low-cost method for measuring dry depth would significantly improve crop yield and quality while conserving water resources. Recent evidence suggests that ultrasonic waves can be a technique to solve this issue. For instance, Chen et al. [32] revealed the correlation mechanism between the law of ultrasonic propagation in coal samples and the migration of water. Tanaka et al. [33] improved the measurement accuracy of soil moisture using an ultrasonic waveguide to predict rainfall-induced slop failure. These studies verified that ultrasonic waves can transmit valuable signals when propagating in substances containing a certain amount of humidity.

Inspired by the aforementioned studies, we have proposed a new method to measure the dry depth of soil using reflective ultrasonic waves. The suitable frequency and propagation time were initially determined through simulation, and the relationship between the reflection time and dry depth was established. Subsequently, an experimental measurement system was implemented to validate the feasibility of the proposed technique. Our findings demonstrate excellent performance, with $R^2$ values exceeding 0.98 and RSEM values below 1.8% for all soil types, surpassing those reported in previous research. This study represents a significant advancement in the field of soil dry depth measurement.

## 2. Materials and Methods

### 2.1. Theoretical Analysis

Soil is a complex and heterogeneous mixture of solid, liquid, and gas components, with pores of various sizes within a solid skeleton. Soil moisture is mainly found in small and medium pores, while large pores are filled with air. Dry soils are defined as those with a moisture content below 5%, as depicted in Figure 1. The depth of dry soil on the surface can be used as a critical point for evaluating drought stress on crops. This measurement takes into account factors such as surface soil water loss and crop absorption, and is expressed in millimeters of depth rather than a water content percentage. It provides a more accurate characterization of soil moisture than traditional water content indices, as it has a clear concept and spatial scale dimension.

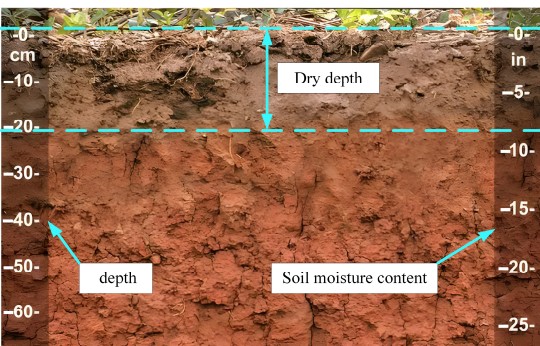

**Figure 1.** Schematic diagram of soil dry depth.

By measuring the depth of the dry soil layer, the critical point between wet and dry soil can be determined, which can help meet crop water needs under specific conditions. The use of soil dry depth in intelligent irrigation systems can improve water management by accurately determining drought stress duration and regulating deficit irrigation parameters. Moreover, the physical quantity of drying depth is practical and relevant to current engineering practices, making it an attractive option for the intelligent manufacturing of agricultural machinery and equipment.

In the process of ultrasonic pulse waves, the distance covered by the wave in one unit of time is referred to as the wave velocity, which can be expressed by:

$$c = \frac{\lambda}{T} = \lambda f \tag{1}$$

where $\lambda$ is the wavelength of the ultrasonic pulse wave, $T$ is the ultrasonic pulse wave period, and $f$ is the ultrasonic pulse frequency.

During the propagation of ultrasonic waves in a medium, the energy is attenuated, leading to the formation of an ultrasonic field. The ultrasonic field is an area of a certain size and shape in which the energy of the ultrasonic pulse wave is distributed. The instantaneous pressure difference at a certain point in the ultrasonic field, with or without ultrasonic energy, is called the sound intensity, which is represented by:

$$P = -\rho c A \omega \sin\left(t - \frac{x}{c}\right) = -P_m \sin\left(t - \frac{x}{c}\right) \tag{2}$$

$$P_m = \rho c A \omega = 2\pi \rho c A f \tag{3}$$

where $P_m$ is the amplitude of sound pressure, $\rho$ is the density of the medium, $c$ is the velocity of ultrasonic wave, and $A$ is the amplitude of the ultrasonic wave. $\omega$ is the angular frequency, $t$ is the time it takes for the ultrasonic wave to be detected, and $x$ is the current displacement.

The velocities of longitudinal and transverse waves are:

$$c_L = \left(\frac{\lambda + 2\mu}{\rho}\right)^{\frac{1}{2}} = \left(\frac{1 - v}{(1 + v)(1 - v)} \times \frac{E}{P}\right)^{\frac{1}{2}} \tag{4}$$

$$c_T = \left(\frac{\mu}{\rho}\right)^{\frac{1}{2}} = \left(\frac{E}{2\rho(1 + v)}\right)^{\frac{1}{2}} \tag{5}$$

where $\mu$ is the Lamé constant related to the elastic properties of the medium, $E$ is acoustic energy, $c_L$ and $c_T$ is wave velocity of longitudinal and transverse waves, and $v$ is the velocity of wave.

It is evident that the wave velocity in a solid medium is solely dependent on its physical properties. Specifically, an increase in the elastic modulus and Poisson's ratio of the medium leads to a decrease in its density, resulting in a reduction of the wave velocity

of an ultrasonic pulse. Furthermore, it is important to note that the longitudinal wave velocity is greater than the transverse wave velocity. Acoustic impedance is the ratio of sound pressure and vibration of a particle in an ultrasonic field, which is represented by:

$$Z = \frac{P}{v} = \frac{\rho c A \omega}{v} = \frac{\rho c v}{v} = \rho c \tag{6}$$

where $P$ is acoustic strength, $\omega$ is angular frequency. $t$ serves as a measure of a medium's resistance to the propagation of sound waves during the ultrasonic pulse process. The value of acoustic impedance is equal to the product of the medium's density and velocity.

Sound intensity is the sum of energy vertically passed by ultrasonic wave in unit time and unit area, and is a physical dimension used to express the intensity of sound wave energy, which is expressed by:

$$I = \frac{1}{2}\rho c A^2 \omega^2 = \frac{1}{2}Z v^2 = \frac{P^2}{2Z} \tag{7}$$

The acoustic pressure and acoustic intensity of the incident wave are $P_0$ and $I_0$, respectively. The sound pressure and intensity of the reflected wave are $P_r$ and $I_r$, respectively. The sound pressure and sound intensity of the projected wave are $P_t$ and $I_t$, respectively. The ratio of $P_r$ to the $P_0$ on the contact surface between the first medium and the second medium is called the sound pressure reflectivity of the contact surface, which is represented by:

$$r = \frac{p_r}{p_0} \tag{8}$$

The ratio of $P_t$ to the $P_0$ is called the sound pressure transmissivity of the contact surface, and is denoted by:

$$t = \frac{p_t}{p_0} \tag{9}$$

It is known that the sound pressure reflectance $r$ and the sound pressure transmittance $t$ are:

$$r = \frac{P_r}{P_0} = \frac{Z_2 - Z_1}{Z_2 + Z_1} \tag{10}$$

$$t = \frac{P_t}{P_0} = \frac{2Z_2}{Z_2 + Z_1} \tag{11}$$

where $Z_1$, $Z_2$ are the acoustic impedances of the first and second media, respectively.

The ratio of $I_r$ to $I_0$ is called the acoustic intensity reflectivity of the contact surface, and is denoted by:

$$R = \frac{I_r}{I_0} = \frac{\frac{P_r^2}{2Z_1}}{\frac{P_0^2}{2Z_2}} = \frac{P_r^2}{P_0^2} = r^2 = \left(\frac{Z_2 - Z_1}{Z_2 + Z_1}\right)^2 \tag{12}$$

The ratio of $I_t$ to $I_0$ is referred to as the sound intensity reflectivity of the contact surface, and is denoted by:

$$T = \frac{I_t}{I_0} = \frac{\frac{P_t^2}{2Z_1}}{\frac{P_0^2}{2Z_2}} = \frac{P_t^2}{P_0^2} = r^2 = \frac{4Z_2 Z_1}{(Z_2 + Z_1)^2} \tag{13}$$

It can be seen from the above formula that when $Z_1 > Z_2$, the acoustic pressure reflectivity $r < 0$, and the transmittance of the second medium is smaller than that of the first medium, part of the incident ultrasonic wave will be reflected back, and part of the incident ultrasonic wave will pass through the contact surface of the two media.

According to the propagation characteristics of ultrasonic waves, when an ultrasonic wave travels from dry soil to wet soil, it encounters a boundary or interface between the

two media. The propagation speed of ultrasonic waves is different in dry and wet soil, and the density of the two media is also different. As a result, the ultrasonic wave has different acoustic impedance in the two media. At the interface between the two media, a reflected wave is generated in the opposite direction of the incident wave, and a transmitted wave travels in the same direction as the incident wave, as shows in Figure 2.

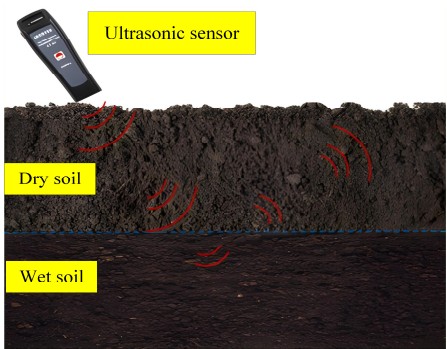

**Figure 2.** Ultrasonic propagation properties in irrigated soil.

As a mechanical elastic wave, ultrasonic energy will be attenuated in the process of propagation along soil medium. This attenuation occurs due to the friction caused by particle vibration and thermal sensing within the medium. In addition to absorption attenuation, the ultrasonic beam will also diffuse in the medium as it propagates, leading to a reduction in ultrasonic intensity within the soil medium.

Figure 3 illustrates the method used for ultrasonic sensing, which involves measuring the ultrasonic reflection wave at the interface between dry and wet soil. This method allows for direct measurement of the depth of dry soil without the need for artificial destruction, as the transmitter and receiver are placed at the same level. This approach is convenient and efficient, as it eliminates errors caused by human disturbance of the soil layer during sensing. Furthermore, the accuracy of the sensing results is significantly improved through this non-invasive approach.

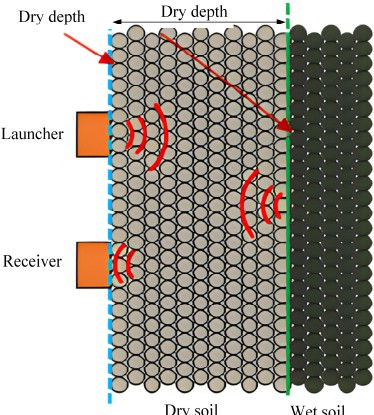

**Figure 3.** The principle of ultrasonic sensor to test dry depth.

## 2.2. Simulation Description

To investigate the relationship between the dry depth of soil and ultrasonic reflection time, the propagation of ultrasonic waves in soil has been simulated in Comsol Multiphysics software via a finite element method (FEM). A 2D soil model with a dimension of 180 mm × 180 mm was created and divided into two layers, as depicted in Figure 4. The left and right sides corresponded to dry soil and wet soil, respectively, and the bisector of the model is set as the dry–wet separation interface. The four sides of the square were set to a low reflection boundary condition by applying an absorbing layer to reduce the non-physical reflections in the physical domain. The dry and wet soil are set with a moisture

content of 5% and 25%, respectively, and the corresponding soil density and sound velocity are listed in Table 1.

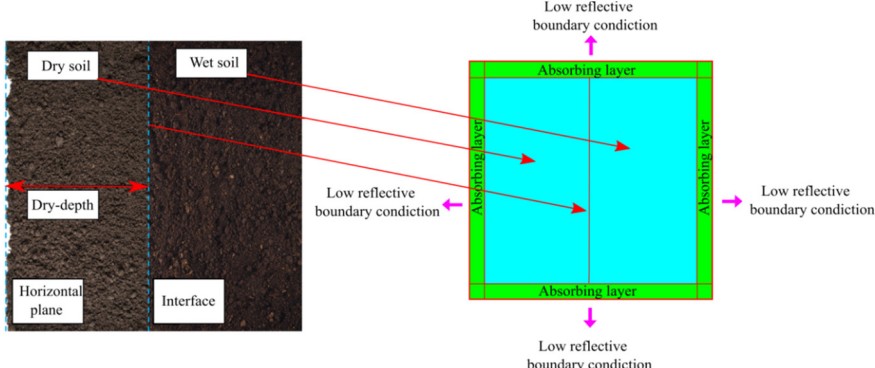

**Figure 4.** The boundary condition imposed for modeling ultrasonic waves propagating in soil.

**Table 1.** Material characteristic parameters of each surface layer of soil.

| Soil Moisture Content | Soil Density (kg/m$^3$) | Sound Velocity (m/s) |
|---|---|---|
| 5% | 1.49 | 520 |
| 25% | 1.67 | 200 |

*2.3. Sample Preparation*

Soil sample preparation was performed at the campus of Guangzhou university (23.030 1.23″ N, 113.240 3.92″ E), and the location is shown in Figure 5. Four different types of soil were prepared, including sandy, sandy loam, clay, and silty loam (See Figure 6). These soils were collected by a standard shovel and then placed in a seal pocket with a label. Forty samples were collected for each soil type and a total of 160 soil samples were prepared. The collected samples were transported to the Solar Energy Intelligent Irrigation Equipment Technology Innovation Center Laboratory of Guangzhou University for processing. All the samples were initially air-dried for 72 h with a light intensity of $1.5 \times 104$ lux and a wind speed of 15 m/s, and then passed through a 2 mm soil sieve to remove gravel and plant roots. The filtered soil samples were placed in rectangular glass tanks, with soil depth set at 60~300 mm. A Dn20 drip irrigation belt with a flow rate of 0.2 L/s was deployed to create different dry depth conditions. After irrigation, the soil was stood for 2.5 h to allow the surface soil moisture to volatilize and sink. A soil-dry-depth measuring system was subsequently established, as depicted in Figure 7. Ultrasonic sensors were placed on the soil surface and an ultrasonic controller was used to collect the data. The wire communication module realized the transmission of the information, and Raspberry Pi was used to for reading and computing sensing data. The practical dry depth of the soil was measured manually using soil moisture sensor (LD-SW, Lynd Intelligent Technology Co., Ltd., Weifang, China).

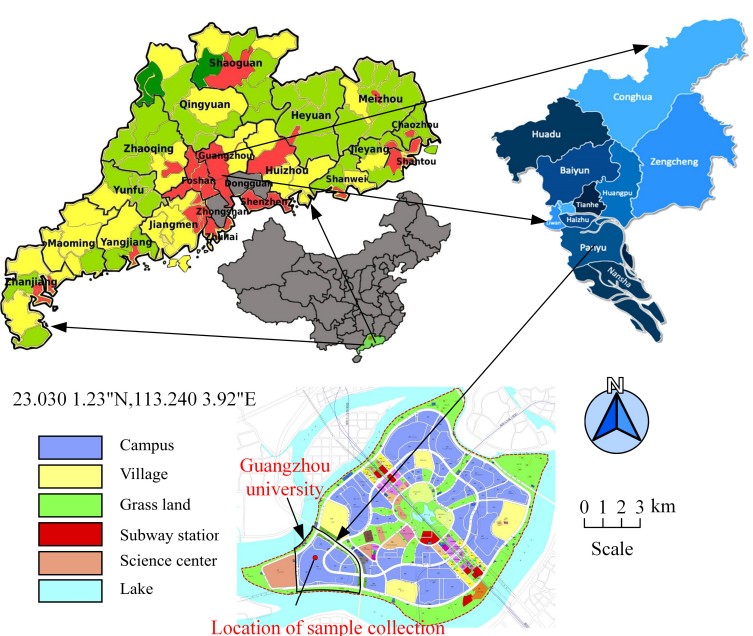

**Figure 5.** The location of Guangzhou university for collecting soil samples.

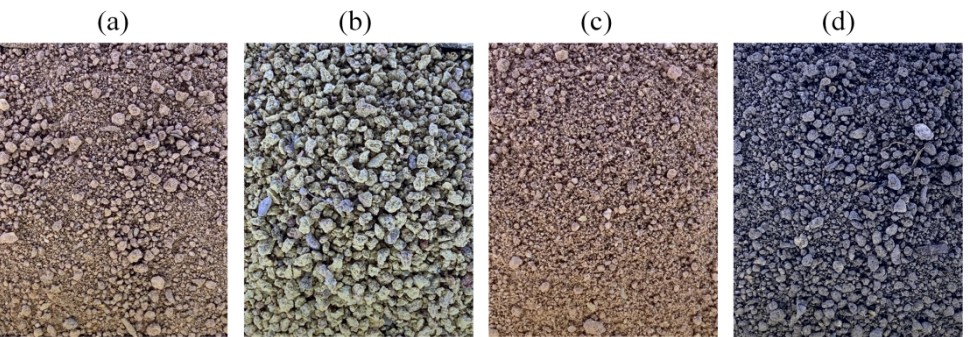

**Figure 6.** Soil samples prepared for experiments: (**a**) clay, (**b**) sandy, (**c**) silty loam, (**d**) sandy loam.

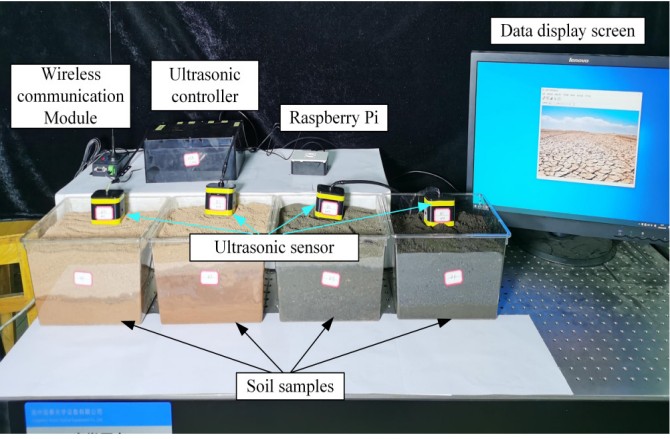

**Figure 7.** The established soil dry depth measuring system.

## 3. Results

Figure 8 reports the propagation velocity of the ultrasonic waves of different frequencies in soil. It was observed that the propagation speed is relatively low at 40 Hz, and the ultrasonic wave cannot reach the interface within 20 μs. As the frequency increases from 40 Hz to 80 Hz, the propagation speed gradually increases. This phenomenon can be explained by the fact that the propagation speed of an ultrasonic wave in the same medium

is proportional to the wavelength and frequency. Typically, the propagation direction of the ultrasonic wave is along the x-axis. However, when an ultrasonic wave encounters an interface between two soil mediums, a portion of it is reflected. This reflection is mainly due to the difference in acoustic impedance between the two mediums.

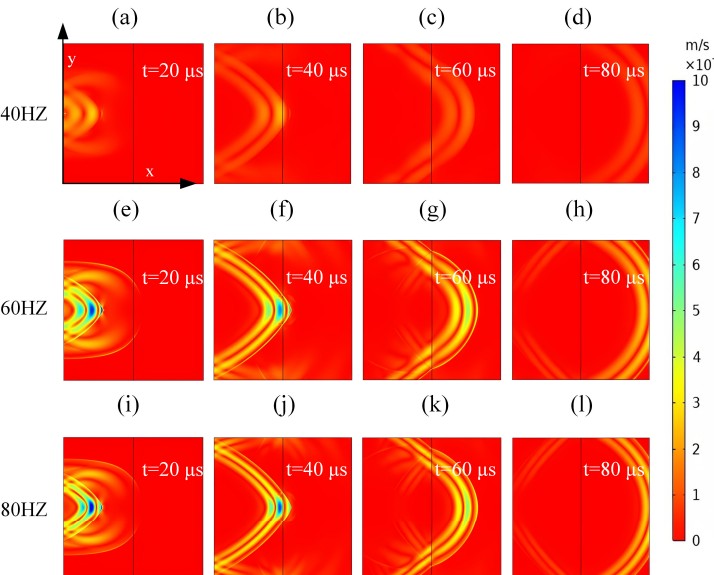

**Figure 8.** The propagation velocity distribution of ultrasonic waves of different frequencies in soil: (**a**–**d**) are the velocity distribution of ultrasonic wave of 40 Hz in the soils with the propagation time of 20 µs, 40 µs, 60 µs, and 80 µs, respectively. (**e**–**h**) are the velocity distribution of ultrasonic wave of 60 Hz in the soils with the propagation time of 20 µs, 40 µs, 60 µs, and 80 µs, respectively. (**i**–**l**) are the velocity distribution of ultrasonic wave of 80 Hz in the soils with the propagation time of 20 µs, 40 µs, 60 µs, and 80 µs, respectively.

Figure 9 reports the pressure distribution of the ultrasonic waves propagating in soil at different frequencies. It was found that the pressure was increased as the propagation time increased from 20 µs to 40 µs. This can be attributed to the fact that the ultrasonic wave passes through two media with different acoustic impedance, leading to the formation of a superimposed wave of direct waves and reflected wave and resulting in an increase in local pressure. A slightly defected wave was observed during the propagation of a second medium, which also can be attributed to the difference in acoustic impedance. As time increased to 60 µs, the ultrasonic wave had almost completely passed through the interface, and the pressure in the second medium dropped significantly. When time reached 80 µs, the pressure in most regions became zero, except for the sound source area of the ultrasonic waves on the left. This is because if the boundary is set as air medium, the ultrasonic wave will slowly decay in the finite space until it disappears, and there will be no second reflection.

In general, although a higher frequency of ultrasonic wave results in a high propagation speed, it also induces a high pressure. To ensure the accuracy of the measurement, the propagation time of the reflected ultrasonic wave in the soil medium is preferably more than 20µs, and a frequency of 60 Hz was selected for these experiments.

Figure 10 illustrates the relationship between reflection time and dry depth for different soil samples, accompanied by a linear fit. It can be seen that the reflection time is almost proportional to the dry depth, especially for sandy loam and silty loam samples, which exhibited high determination coefficients of 98.24% and 97.4%, respectively. Notably, for a fixed dry depth, the reflection time of clay is comparatively lower than that of sandy loam, silty loam, and sand samples. This can be attributed to its fine particles, numerous pores, and weak water permeability, which contribute to an accelerated propagation speed of ultrasonic waves.

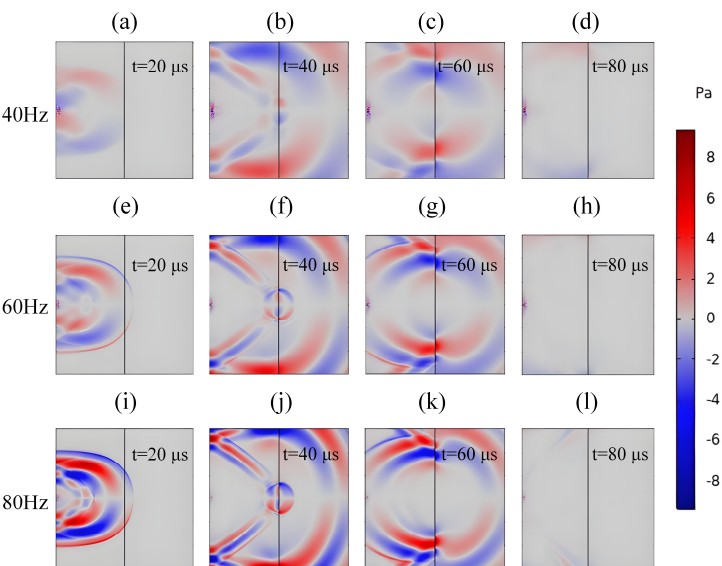

**Figure 9.** The propagation pressure distribution of ultrasonic waves of different frequencies in soil: (**a**–**d**) are the distribution of ultrasonic wave of 40 Hz in the soils with the propagation time of 20 μs, 40 μs, 60 μs, and 80 μs, respectively. (**e**–**h**) are the distribution of ultrasonic wave of 60 Hz in the soils with the propagation time of 20 μs, 40 μs, 60 μs, and 80 μs, respectively. (**i**–**l**) are the distribution of ultrasonic wave of 80 Hz in the soils with the propagation time of 20 μs, 40 μs, 60 μs, and 80 μs, respectively.

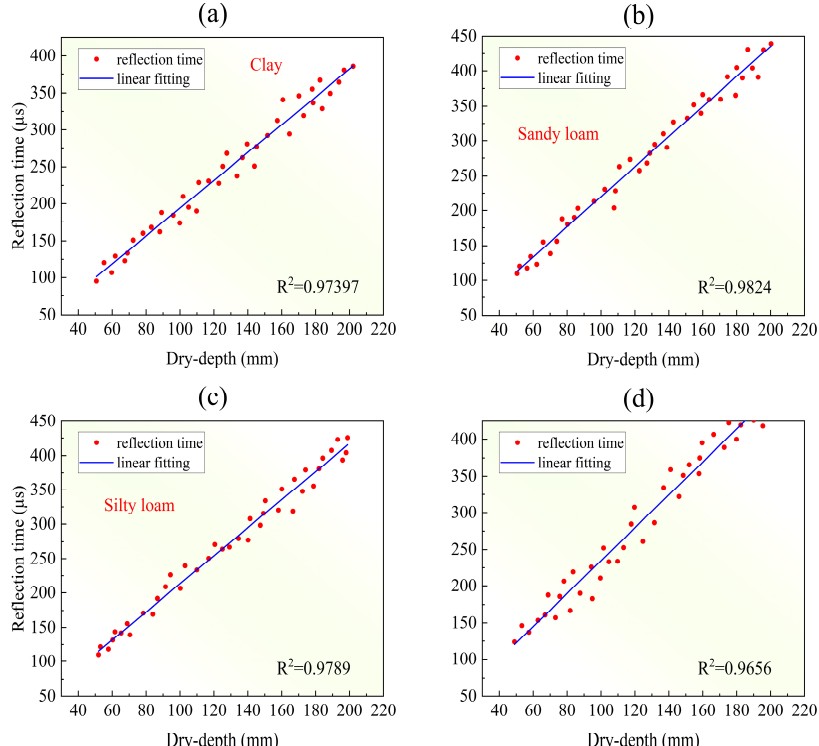

**Figure 10.** The relationship between reflection time (μs) and dry depth (mm) for (**a**) clay, (**b**) sandy loam, (**c**) silty loam, and (**d**) sand, respectively.

Figure 11 shows the error analysis between the ultrasonic and manual measurement of dry depth for different soil samples. It was found that the ultrasonically and manually measured values reach a great agreement for the sandy loam and silty loam samples with only a minor discrepancy observed in the regression model. The corresponding $R^2$ values

are 0.9899 and 0.0990, and the RMSE values are 0.01572 and 0.01599, respectively. Although a high accuracy was also obtained for sandy loam ($R^2$ values of 0.09896, RMSE value of 0.0166) and silty loam ($R^2$ values of 0.09874, RMSE value of 0.0177) samples, significant differences were observed for clay and sand samples in certain measurements. Specifically, errors were prominent in the fifth, eleventh, and fourteenth measurements for clay samples and the thirty-fifth and thirty-sixth measurements for sandy samples. The effectiveness of the proposed method can vary across different soil types, which is attributed to variation in the physical properties of soil. The speed of an ultrasonic wave through soil depends on its density and elasticity, which are impacted by texture and compaction. These differences can cause the wave to scatter or to be absorbed differently, resulting in variations in the quality of the reflected signal. In clay and sandy soils, ultrasonic waves are subject to more scattering and attenuation when propagating inside the soil due to the tightly packed arrangement of particles and relatively low porosity. As a result, the reflected signals are weak. In contrast, sandy loam and silt loam soils have larger soil particles, higher porosity, and a lower soil density, which allows ultrasonic waves to propagate in a more direct and clear path within the soil. This leads to relatively stronger reflected signals. In addition, since sandy and clay soil contain a high content of sand, the water in these soils evaporates easily, resulting in higher ultrasonic measurement values compared with manual measurements.

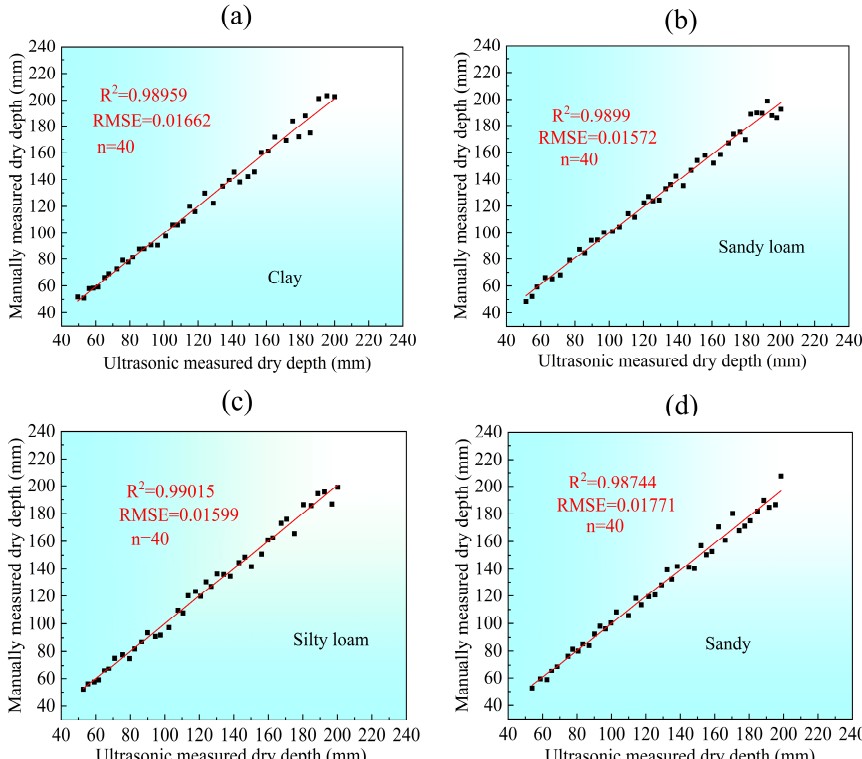

**Figure 11.** The error analysis between ultrasonically and manually measured dry depth for (**a**) clay, (**b**) sandy loam, (**c**) silty loam and (**d**) sandy, respectively.

The maximum measurement errors for each soil sample is listed in Table 2. The maximum measurement error in sandy loam and sandy is larger than in the silt and clay samples with values of 5.97% and 5.98, respectively, while the maximum measurement error of clay and silty loam is 5.67% and 5.89, respectively. Despite these measurement errors, the overall accuracy is acceptable. These findings demonstrated the feasibility of using ultrasonic waves to measure the dry depth of the soil.

**Table 2.** The maximum measurement error for different soil samples.

| Soil Types | Manually Measured | Ultrasonic Measured | The Maximum Error |
|---|---|---|---|
| Clay | 96.23 mm | 90.78 mm | 5.67% |
| Sandy loam | 197.85 mm | 186.05 mm | 5.97% |
| Silty loam | 120.19 mm | 113.51 mm | 5.89% |
| Sandy | 62.54 mm | 58.8 mm | 5.98% |

## 4. Discussion

Table 3 presents a comparison of the measurement accuracy of the proposed method with other techniques. TDR and capacitance sensor show a good performance with an $R^2$ value > 0.87, but the maximum RMSE values exceed 2%. The neutron probe method exhibits some uncertainties, with RMSE values ranging from 2%t to 6.7%. The electrical resistivity method shows relatively low $R^2$ values (i.e., 0.21~0.68), although a lower RMSE value can be realized (i.e., 1.15%). In contrast, the proposed method in this work achieved $R^2$ values of over 0.98 and RMSE values less than 1.8%. This outstanding performance further demonstrates the potential of the proposed method for water management, soil remediation, and precision farming. Moreover, the non-invasive and low-cost implementation make it an attractive option for field applications.

**Table 3.** The accuracy of the proposed method compared with the techniques previously reported.

| | Techniques | $R^2$ | RMSE |
|---|---|---|---|
| This work | Reflected ultrasonic wave | 0.98~0.99 | 1.5%~1.7% |
| Huisman et al. [34] | TDR | 0.87~0.99 | 1.4%~4.4% |
| Robinson et al. [35] | Capacitance sensors | 0.88~0.94 | 1.4%~2.4% |
| Ochsner et al. [36] | Neutron probe | 0.74~0.99 | 2%~6.7% |
| Cao et al. [37] | Carbon-fiber heated cable | 0.72 | 2.85% |
| Calamita et al. [24] | Electrical resistivity | 0.21~0.68 | 1.15%~6.03% |
| Zhao et al. [23] | L-band Radiometry | Not mentioned | 4.1% |

However, there are limitations to the proposed method. First, the measurement accuracy could be impacted by soil moisture of dry oil. In extremely dry soil with less than 1% moisture, the lack of moisture can create a relatively uniform density throughout the soil, making it difficult for the ultrasonic waves to reflect back to the surface. This can lead to inaccurate readings or an inability to measure the dry depth of the soil at all. Second, soil type can affect the accuracy of ultrasonic reflective wave measurements, with high levels of clay or organic matter scattering ultrasonic waves, and making it difficult to obtain accurate readings.

It is worth noting that this study only considered four soil types and carried out measurements at room temperature, while many factors, such as wind, temperature, and sunlight were not taken into account. Future research should aim to improve the applicability of this method by considering all soil types and accounting for the effect of environmental factors.

## 5. Conclusions

In this study, a method based on a reflective ultrasonic wave was proposed to measure the dry depth of soil. A theoretical investigation was initially conducted to demonstrate the feasibility of such a method. A frequency of 60 Hz and a suitable propagation time of greater than 20 μs was selected. Then, an experimental measurement system was established to validate the consistency between ultrasonic measurements and manually measured values. The results show that the ultrasonic measurements show a great agreement with the manual measurements. The resulted $R^2$ values and RSME values are over 0.98 and below 0.3%, respectively, which is better than some of the methods previously reported. Our findings provide evidence that the proposed ultrasonic measurement could be an efficient

and low-cost method for measuring soil dry depth, which helps to improve the usage of water resources.

**Author Contributions:** Conceptualization and methodology, J.X.; software, validation, and formal analysis, S.L.; resources, data curation, and visualization, C.Z.; investigation and writing—original draft preparation, Z.L.; writing—review and editing, Y.Z.; supervision, project administration, and funding acquisition, Z.Z. All authors have read and agreed to the published version of the manuscript.

**Funding:** This research was funded by National Natural Science Foundation of China (51975136, 52075109), the Science and Technology Innovative Research Team Program in Higher Educational Universities of Guangdong Province (2017KCXTD025), the Industry-University-Research Collaborative Innovation Base of Ministry of Education (220903950010408), Special Research Projects in the Key Fields of Guangdong Higher Educational Universities (2019KZDZX1009), Natural Science Foundation of Guangdong Province (2023A1515011723), the Tertiary Education Scientific research project of Guangzhou Municipal Education Bureau (202235139), and Guangzhou University Research Project (YJ2021002).

**Data Availability Statement:** The data that support the findings of this study are available from the corresponding author upon reasonable request.

**Acknowledgments:** We thank the editors for their hard work and the referees for their comments and valuable suggestions that helped to improve this paper.

**Conflicts of Interest:** The authors declare no conflict of interest.

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
