# Peer review of "Accurate and Rapid Measurement of Soil Dry Depth Using Ultrasonic Reflection Waves"

_agronomy, doi:10.3390/agronomy13051276_

Round 1
Reviewer 1 Report
Comments are attached.

Author Response
A point-by-point response has been added in the attached PDF file

Reviewer 2 Report
The manuscript needs to go through some minor revision before publishing. The points to be cleared or adjusted are listed below:
1. Introduction:
The last sentece sounds like to be in the abstract. Here should sound like the objetive of the research.
Surprisingly, the results obtained from ultrasonic measurements were in great agreement with those manually measured dry depth values. Although some errors were observed in measuring the dry depth of certain types of soil, the overall accuracy was acceptable. These findings show that the proposed method could be an efficient and cost-effective approach to measuring the dry depth of soil.
2. Materials and Methods:
Eq.3 - Define what is Õ¡, t and x;
Eq.4 - Define what is v;
Line 207: what type of soil moisture sensor was used?;
Line 245 and 246: unit correction: µs not us;
Fig.9: unit correction for reflection time in all figures; Fig. Caption: The relationship between reflection time (UNIT?) and dry depth (UNIT?) for (a) clay, (b) sandy loam, (c) silty loam, and (d) sand, respectively;
Fig. 10. I Expected to see a linear regression comparison as Fig.9, between measured manually (x-axis) and with the instrument (y-axis) for each soil type
Author Response
a point-by-point response has been added in the attached PDF file
